# Guidelines on Standard and Therapeutic Diets for Adults in Hospitals by the French Association of Nutritionist Dieticians (AFDN) and the French Speaking Society of Clinical Nutrition and Metabolism (SFNCM)

**DOI:** 10.3390/nu13072434

**Published:** 2021-07-15

**Authors:** Marie-France Vaillant, Maud Alligier, Nadine Baclet, Julie Capelle, Marie-Paule Dousseaux, Evelyne Eyraud, Philippe Fayemendy, Nicolas Flori, Esther Guex, Véronique Hennequin, Florence Lavandier, Caroline Martineau, Marie-Christine Morin, Fady Mokaddem, Isabelle Parmentier, Florence Rossi-Pacini, Gaëlle Soriano, Elisabeth Verdier, Gilbert Zeanandin, Didier Quilliot

**Affiliations:** 1Service Diététique, CHU Grenoble Alpes, CS 10217, CEDEX 9, 38043 Grenoble, France; mfvaillant@chu-grenoble.fr; 2Laboratoire de Bioénergétique Fondamentale et Appliquée, Université Grenoble Alpes, U1055, CS 40700, CEDEX 9, 38058 Grenoble, France; 3FORCE (French Obesity Research Center of Excellence), FCRIN (French Clinical Research Infrastructure Network), CRNH Rhône-Alpes, Centre Hospitalier Lyon Sud, 165 Chemin du Grand Revoyet, 69310 Pierre-Bénite, France; maud.alligier@chu-lyon.fr; 4Service Diététique, Pitié Salpêtrière, Assistance Publique-Hôpitaux de Paris, 47-83, Bd de l’Hôpital, CEDEX 13, 75651 Paris, France; nadine.baclet@wanadoo.fr (N.B.); mariepaule.dousseaux@free.fr (M.-P.D.); 5Service Diététique, Centre Hospitalier Simone Veil de Blois, Mail Pierre Charlot, 41000 Blois, France; capellj@ch-blois.fr; 6Service Diététique, CHU de Nice Hôpital de l’Archet, 151 Route Saint Antoine de Ginestière, 06200 Nice, France; eyraud.e@chu-nice.fr; 7Unité de Nutrition, CHU Dupuytren, 2, Avenue Martin-Luther-King, CEDEX, 87042 Limoges, France; Philippe.Fayemendy@chu-limoges.fr; 8UMR 1094 Inserm Associée IRD—Neuroépidémiologie Tropicale, Faculté de Médecine, 2, Rue du Docteur Marcland, CEDEX, 87025 Limoges, France; 9Clinical Nutrition, Gastroenterology and Endoscopy, Institut Régional du Cancer Montpellier (ICM), University of Montpellier, Parc Euromédecine, 208 Rue des Apothicaires, 34298 Montpellier, France; Nicolas.Flori@icm.unicancer.fr; 10Nutrition Clinique, Service d’Endocrinologie-Diabétologie-Métabolisme, Centre Hospitalier et Universitaire Vaudois, 1011 Lausanne, Switzerland; Esther.Guex@chuv.ch; 11RESCLAN Champagne-Ardenne, Hôpital Sébastopol, 48, Rue de Sébastopol, 51092 Reims, France; resclan@orange.fr; 12Service Diététique, Centre Hospitalier Régional Universitaire de Tours, CEDEX 9, 37044 Tours, France; F.Lavandier@chu-tours.fr; 13Unité Diététique, Hôpital Larrey, CHU de Toulouse, 20, Av. Larrieu-Thibaud, 31100 Toulouse, France; martineau.caroline@chu-toulouse.fr; 14Service Diététique, Assistance Publique Hôpitaux de Marseille, Chemin des Bourrely, CEDEX 20, 13915 Marseille, France; MarieChristine.Morin@ap-hm.fr; 15Service de Gastro-Entérologie, Cliniques Sud Luxembourg Vivalia, Rue des Déportés 137, 6700 Arlon, Belgium; fady.mokaddem@skynet.be; 16Service Diététique, CHRU Lille, 2 Avenue Oscar Lambret, 59037 Lille, France; iparmentier@chru-lille.fr; 17Coordination Générale des Soins, Assistance Publique–Hôpitaux de Marseille, 80, Rue Brochier, CEDEX 05, 13354 Marseille, France; florence.pacini@ap-hm.fr; 18Gérontopôle, CHU Toulouse, CEDEX 9, 31059 Toulouse, France; gaelle.soriano.diet@gmail.com; 19Service diététique, Hospices Civils de Lyon, Hôpital Femme Mère Enfant, 59, Bd Pinel, CEDEX, 69677 Bron, France; elisabeth.verdier@gmail.com; 20Cabinet des Maladies de l’Appareil Digestif et Nutrition Clinique, Palais Bel Canto, 29, Avenue Malaussena, 06000 Nice, France; gilbert.zeanandin@gmail.com; 21Unité Transversale de Nutrition et Unité d’Assistance Nutritionnelle, Service d’Endocrinologie Diabétologie et Nutrition, CHRU de Nancy, Rue du Morvan, 54500 Vandoeuvre-lès-Nancy, France

**Keywords:** food, therapeutic diet, prescription, hospital, guidelines

## Abstract

Aim: Hospital food provision is subject to multiple constraints (meal production, organization, health safety, environmental respect) which influence the meal tray offered to the patient. Multiple diets can add complexity and contribute to non-consumption of the meal. To avoid undernutrition, it appeared necessary to propose guidelines for foods and diets in hospitals. Methods: These guidelines were developed using the Delphi method, as recommended by the HAS (French Health Authority), based on a formal consensus of experts and led by a group of practitioners and dieticians from the AFDN (French Association of Nutritionist Dieticians) and SFNCM (French Society of Clinical Nutrition and Metabolism). Results: Twenty-three recommendations were deemed appropriate and validated by a panel of 50 national experts, following three rounds of consultations, modifications and final strong agreement. These recommendations aim to define in adults: 1—harmonized vocabulary related to food and diets in hospitals; 2—quantitative and qualitative food propositions; 3—nutritional prescriptions; 4—diet patterns and patient adaptations; 5—streamlining of restrictions to reduce unnecessary diets and without scientific evidence; 6—emphasizing the place of an enriched and adapted diet for at-risk and malnourished patients. Conclusion: These guidelines will enable catering services and health-care teams to rationalize hospital food and therapeutic food prescriptions in order to focus on individual needs and tasty foods. All efforts should be made to create meals that follow these recommendations while promoting the taste quality of the dishes and their presentation such that the patient rediscovers the pleasure of eating in the hospital.

## 1. Introduction

### 1.1. Hospital Food Provision: Interest and Societal Challenges

While medical knowledge, therapeutic molecules and treatment avenues have evolved, disease-related diets have more to do with (a long-standing tradition of prescription, long-standing prescribing habits). The questioning of these practices must be part of a rational approach based on evidence-based medicine and, failing the latter, on expert opinions.

### 1.2. Constraints Related to Meals

Food provision in the hospital setting encompasses multiple dimensions: biological, economic and health security, but also socio-cultural, symbolic and ecological. Faced with such challenges, food provision for the patient is a complex act insofar as it is at the very end of the nutritional chain where catering staff, caregivers and hospital management intervene directly or indirectly on the meals that are distributed to the patients. Food catering services have evolved over the course of many decades [1], notably taking into account both the economic and organizational aspects (invention of the cook-and-chill method, meal recovery carts, food waste, etc.). Meal quality mobilizes the various catering and nutritional teams, as well as hospital management. New food trends are also influencing the desires of consumers who demand the latter after their hospitalization. Nevertheless, the hospital meal remains negatively connoted by professionals, as well as by patients, as highlighted by results of satisfaction surveys (i.e., the e-Satis survey proposed by the French Health Authority (HAS, Haute Autorité de Santé)).

### 1.3. Nourishing the Ill Patients

The above ensemble of facts determines and influences hospital meals, in which nutritional diets occupy a predominant place. Thus, diet prescriptions as well as proposals (types of dishes, preparation methods, possible restrictions such as salt, fat and carbohydrates in the recipe, etc.) can lead to a risk of non-consumption contributing to the spiral of undernutrition [2].

### 1.4. Objectives of the Present Guidelines

The objective of this work is to propose recommendations regarding food and nutritional management and prescriptions based on scientific evidence, or failing the latter, on expert consensus.

The main objectives of these recommendations are:To ensure the nutritional needs of the patients and adapt to their pathophysiological condition as well as to the needs associated with hospitalization (acute care, follow-up and rehabilitation care, long-term stay).To show consideration for the patient by conferring food provision its full meaning: both nutrition-wise as well as personal-wise (taking into account choices, mealtimes, etc.).To rationalize and harmonize practices related to nutritional diet prescriptions.To lighten the constraints that could induce restrictions on meal provision (choice of food for the preparation of dishes, implications on meal costs, consequences on the palatability and consumption of meals/dishes by patients).

These recommendations are intended for the personnel involved at every level of the food chain, including people involved in patient care: catering industry overseers (catering engineer or kitchen manager), catering assistants, care workers; nursing staff supervisors: managers, senior health managers and care coordinators; dieticians, nurses, caregivers, hospital services agents, physicians, upper management overseers such as hospital heads, purchasing or economic services directors/logistics directors, and institutional stakeholders.

## 2. Materials and Methods

### 2.1. Description of the Method

The present guidelines were developed using the following methodology:A preliminary survey was carried out by the French speaking Society of Clinical Nutrition and Metabolism (SFNCM, Société Francophone de Nutrition Clinique et Métabolisme) in 2017, with the goal of establishing an inventory of food practices and diets in French healthcare establishments [3]. This survey revealed a wide heterogeneity in the various nutritional regimens offered, whereby the same diet name could correspond to different contents in terms of authorized or prohibited foods as well as threshold levels in the case of restrictions. This discordance, induced by historical practices or prescribing habits, has direct consequences on the food intake of patients and their nutritional status.This inventory led to the creation of a working group comprised equally of members of the French Association of Nutritionist Dieticians (AFDN, Association Française des Diététiciens Nutritionnistes) and the SFNCM. Proposals for recommendations were put forward based on the results of the survey and data from the literature.These proposals were submitted to a group of 50 national experts (25 dieticians and 25 physicians). The experts were drawn from all regions of the country and chosen for their expertise in the various specialties pertaining to nutrition (gastroenterology, nephrology, obesity, etc.). The experts were consulted on the basis of the DELPHI method, as proposed by the French Health Authority (HAS) [4].

### 2.2. Procedural Design of the DELPHI Method

The DELPHI method was used to obtain formal consensus on the recommendations. This steering group (14 clinical dieticians, 5 physician nutritionists, 1 methodology assistant) henceforth:Analyzed, synthesized and debated the literature on hospital food provision and nutritional diets.Produced 22 initial proposals stemming from the baseline survey of 2017.Submitted the 22 version 1 recommendations online for a 1st assessment rating (score ranging from 0 to 9) by the 50 experts.Analyzed the responses from this first round: the comments were used to upgrade the proposals and ultimately led to proposing a 23rd recommendation to complete version 1.Submitted the 23rd recommendation as well as the version 2 recommendations (with accompanying arguments) for a second assessment rating of the version 1 proposals which had an initial rating below 7.Analyzed the responses of the 2nd round. All of the recommendations were deemed appropriate (median ≥ 7) with strong agreement for 18 of the latter (scores between 7 and 9), and relative agreement (scores between 5 and 9) for the remaining 5. The experts who gave a score less than 7 on these 5 recommendations with relative agreement were contacted individually for a 3rd rating and comments.Following the 3rd round, the 23 recommendations were all accepted with strong agreement.

All proposals were the subject of a drafting of a supporting rationale based on a comprehensive analysis of the literature and submitted to the experts during the 2nd round. Certain recommendations are based on a consensus of experts in instances where literature data are non-existent or insufficient.

### 2.3. The 23 Recommendations on Standard and Therapeutic Diets

#### 2.3.1. Lexical Scope of the Recommendations

Given the disparity of terminologies used in health establishments [3], it appeared essential for the steering group to redefine and harmonize the various terminologies.

Whether in acute care or rehabilitation, whether in health establishments or in-city facilities, each stakeholder involved in the care of the patient must convey the same level of care management information, both in terms of quality (indication of the prescription, foods or nutrients involved) and quantity, etc.

In addition, for the purpose of eliminating directly identifying designations, at times including the patient’s diagnosis via the meal sheet, the decisions were to:Prefer the term “standard” to “normal” with regard to standard food offered in health establishments.Only retain therapeutic diets—i.e., that imply restrictions or modifications on foods—that fall under prescribed medical settings.

All of these proposed changes are intended to provide a vocabulary more suited to the evolution of care management and knowledge. In the recommendations relative to the gastrointestinal field and fiber control component, vocabulary development choices related to current knowledge on the topic are notably specified.

#### 2.3.2. Structuring of the Recommendations

The recommendations aim to:Define food provision (quantitative and qualitative) in hospital establishments, both in terms of adaptation to the patient’s needs as well as to promote food intake (recommendations 1, 2, 3, 4).Highlight the indications/importance of nutritional prescription and its re-assessment (recommendations 5, 6, 7).Specify the food provision modalities that must sometimes be implemented in order to adapt the diet to the patient’s capabilities and appetite (recommendations 8, 9).Streamline and limit the restrictions in therapeutic diets which exclude certain nutrients, in order to only retain the indications based on scientific evidence and to define these indications with regard to the benefit/risk balance of undernutrition (recommendations 10 to 22).Highlight the importance of the place of specifically adapted diets for undernourished patients and those at risk of undernutrition (recommendation 23).

Scope of the recommendations: These recommendations are only intended for adult populations, pediatrics being a very specific field which requires completely distinct recommendations.

## 3. Results

### 3.1. Recommendations on Food Provision

**Recommendation 1**: It is recommended that food provision offer several choices to the patient.

Comment:

Qualitative and quantitative food provision should be considered an essential component of the treatment of hospitalized patients [5,6]. Two recent cross-sectional Nutrition Day^®^ surveys conducted in nearly 20,000 patients in European and Australian hospitals demonstrated the reality of nutritional risk and highlighted that nearly one in two patients consumed less than half of the meals served [7,8]. Dupertuis et al. in 2003 and Thibault et al. in 2011 furthermore described an association between protein-energy deficits, lower evening meal quality and specific diet prescriptions, suggesting the importance of the quality of food provision [9,10]. Inadequate food provision, without the possibility of choosing menus, more easily exposes the patient to his/her dislikes, or to a meal that does not integrate his/her culinary preferences, lifestyle, beliefs, culture, or philosophy of life [11,12].

The objective of food provision of a health establishment (HE) is to ensure the nutritional needs of each patient [6]. This goal can only be achieved if the patient’s tastes, dislikes, requests and clinical condition are taken into account [13]. Thus, the hospital should offer users a tangible possibility of personal choice [6,14], by making available a list of foods that is as broad as possible, taking into account the variable constraints according to the HE [13,15]. This choice of menus must allow the fulfilling of all nutritional diets (standard or therapeutic), in accordance with the recommendations, and taking into account allergies, dislikes and/or patient preferences [6].

Few studies have examined the impact of choice of menu preferences. In a 2013 study by Jarrin et al., choosing the meal in an internal medicine department did not affect the satisfaction felt by patients and did not increase their food intake [16]. However, this single-center observational study was carried out in a single short-stay department, on a small number of patients with acute disease, and hence not a reflection of all hospitalized patients. According to other authors, provision of food comprising preferred menu choices with the possibility of indicating food dislikes would limit patient dissatisfaction, a frequent source of lower consumption and risk of undernutrition [17]. Accordingly, for Stanga et al., patient satisfaction was improved by the choice of menu preferences [18]. According to Mosqueira et al. in 1996 and Folio et al. in 1998, the proposing of multiple-choice menus, ordered at bedside, within the limits of available choices in the HE, improved patient satisfaction regarding their meal [19,20]. Other authors, however, suggest that menu choices left to the patient alone are not necessarily beneficial if, for example, the most undernourished individuals choose a low-energy menu, normally adapted to a healthy person: menu choices, and particularly in instances of implementation of a therapeutic diet, should probably be supervised and guided according to the patient’s clinical condition [21]. Finally, the possibility of preferred menu choices appears to be an effective means to significantly reduce food waste [17,22,23].

**Recommendation 2**: It is recommended that the standard diet meet the guidelines for prevention and health promotion.

Comment:

Food provision in health establishments (HE) must enable fulfilling the needs for maintaining a proper nutritional and hydration state while respecting a balanced diet as much as possible [13].

There are no data to date comparing the average nutritional needs of hospitalized individuals with those of the general population. It is likely that a standard diet, based on recommendations for the prevention and promotion of health in the general population, does not meet the needs of many hospitalized patients, e.g., undernourished patients or those at risk of undernutrition, and patients with metabolic syndrome or overload. Therefore, an initial nutritional assessment and subsequent adaptation of the standard diet into a therapeutic diet tailored to the patient’s clinical condition is imperative.

Nevertheless, a standard diet corresponds to the basic food provision often proposed by default to the greatest number of hospitalized individuals. In addition, several reports condemn the excessive use of therapeutic diets, in particular restrictive diets [6,14], emphasizing the need for a more frequent use of a healthy, balanced and appropriate standard diet. In order to propose a framework for HEs, it is, therefore, necessary to provide a consensus definition of standard diet. Subject to the constraints of collective catering, standard nutrition must enable meeting the needs of the greatest number of hospitalized individuals who do not require a therapeutic diet. In addition, due to the need for both educational and pedagogical actions in the nutritional sphere of patients and their family members, as recommended by the High Authority of Health, the French High Council of Public Health and the National Food Council [6,24,25], it would hence seem appropriate that this standard diet be offered in coherence with the nutritional policy of prevention and health promotion.

There have been, for several years, recommendations regarding nutrition from a public health perspective. In France, the main example is the National Nutrition Health Program (PNNS, Programme National Nutrition Santé) launched in 2001 and its recommendations for a healthy and balanced diet [25,26]. As a result, the High Council of Public Health has defined a nutritional policy compiling objectives of food consumption, nutritional intake, and nutritional status for the general population as well as for hospitalized patients [25]. The implementation of a standard diet in HEs meeting consumption benchmarks defined for the general population would appear to be consistent with the decrease in the prevalence of nutritional disorders (undernutrition and/or overweight) [6,25]. Whereas the PNNS recommends a guideline, the proposal of a standard, balanced and varied diet meeting nutritional benchmarks requires compliance with precise specifications in terms of quality, frequency and acceptability, while also meeting the constraints of collective catering. The nutrition recommendations of the Public Catering and Nutrition Markets Study Group (GEM-RCN, Groupe d’Etude des Marchés de Restauration Collective et Nutrition) allow HEs to define and apply such specifications [26]. Standard food provision, offered in HEs in accordance with the recommendations of the GEM-RCN, favors a balanced and varied diet. The objective of this provision is to respond to the increase in the prevalence of overweight and obesity, but also to counter undernutrition in elderly and/or dependent individuals [27].

**Recommendation 3**: Outside of a personalized adaptation, it is recommended that the standard diet provide a minimum of 2000 kcal/d.

Comment:

The nutritional needs of the patient must be established according to his/her weight, metabolic condition and appetite. However, the majority of current meal distribution systems do not allow meal trays to be personalized by adapting the size of the portions served. A simple determination of nutritional requirements by applying 30 kcal/kg/day (even for the elderly, grade B [27]) makes it possible to establish the energy target of a nutritional day. In order to fulfill the nutritional needs of a large number of patients, we recommend a minimum of 2000 kcal per day provided by meals. This threshold takes into account the characteristics of the hospitalized population. The Nutrition-Day survey [7] of 15,123 hospitalized patients revealed a median weight of 70 kg. Data from 17,102 screenings in 2017 and 2018 in the medical and surgical departments of a French University Hospital Center showed very constant monthly figures, with a median of 68.2 kg and a mean of 72.4 ± 9.9 kg, respectively (unpublished data). The 2000 kcal cover the needs of half of the patients. They represent a minimum daily threshold to be provided, given that meals are not always fully consumed and, depending on the proposed menus, there can also be significant energy content differences from one day to the next.

The issue of food waste [23,28] must also be taken into account when proposing strategies to improve food intake. These include the conditions of meal intake, an appetizing presentation of the dishes, the quality in the choice of foodstuffs, and the palatability of the recipes. There may also be strategies aimed at improving the nutrient density of the dishes, without increasing their volume. Snacks can also be used to reach nutritional targets, although one must ensure that they are systematically distributed to the patients.

**Recommendation 4**: It is recommended that the nutritional values (energy, proteins, fats, carbohydrates) of provided food be accessible, in particular to the prescribing physician and the dietician.

Comment:

Knowledge of the nutritional values of the provided food is a prerequisite that allows the assessment of the actual provision offered to patients and guarantees its regularity. The Food and Nutrition Liaison Committee (CLAN, Comité de Liaison Alimentation Nutrition) [29] and the dietetic service, or in the absence of such structures, the dietician of the health establishment must validate the provided foods in order to facilitate the work of the prescribers. These prescribers will hence have knowledge of the overall characteristics of the provided food (energy, proteins, etc.), in order to tailor nutritional provision to specific situations, when necessary. For therapeutic diets, the nutritional values of these dishes may target salt, fiber, potassium intake, etc., depending on the purpose of the prescription. Knowing these values allows the assessment of the patient’s true intake in order to ascertain whether the prescribed diet fulfills the patient’s nutritional needs and in what proportions, in order to recommend a readjustment or, if needed, nutritional support [30,31].

The estimated nutritional value must be calculated:For dishes prepared by catering services: knowledge per serving served, at minimum—proteins, lipids, carbohydrates and energy.For therapeutic diets, the nutritional values of these dishes can target salt, fiber, potassium, calcium intake, etc.For prepared products originating directly from the food industry, suppliers must be able to provide the nutritional value of the dishes.Special attention is required for each change in supplier, for both finished dishes and raw foodstuffs. An update of the calculations is then necessary.Given that the calculations are based on recipe technical sheets as well as technical sheets of raw foodstuffs from suppliers, approximations are frequent.Regular nutritional analyses by appropriate analysis laboratories are to be integrated into the follow-up and quality control of the catering service. They are the guarantors of the regularity and adequacy of food provision along with the nutritional target. They also allow strategies to be adjusted in terms of menus, palatability, and the reaching of nutritional goals.

### 3.2. Prescription of Nutritional Diets at the Hospital

**Recommendation 5**: It is recommended that all diets (standard and therapeutic) be prescribed upon admission of the patient and adapted according to his or her clinical condition.

Comment:

The prescription includes all nutritional diets: standard and therapeutic (restrictive, fortified therapeutic diets and/or texture modified diets).

Many authors agree that the provision of meals should be considered an essential component of patient treatment [5,29]. The provision of meals upon admission, tailored to the patient’s clinical condition, is a broad-based process involving many stakeholders in healthcare establishments (physicians, dieticians, caregivers and catering services). This process begins with the identification of the patient’s nutritional status, nutritional risk, food intake possibilities, and the definition, as specific as possible, of his/her clinical condition [6,32].

The act of eating, i.e., the non-contraindication to food intake, and the type of diet must be subject to a medical prescription [6,33]. The nutritional prescription is a medical act which implies the physician’s medico-legal responsibility [33]. The physician prescribes the patient’s nutritional diet [6], standard or therapeutic (including texture modified), and determines the implementation of specific measures in the event of inability, insufficiency or contraindication of an oral diet.

While the responsibility for the nutritional prescription rests with the physician, the tailoring of the diet at admission can be subject to the expertise of a dietician [6,13,34,35]. The training of physicians in the field of nutrition is very heterogeneous and often of average quality [6]. In addition, not all health establishments include a physician nutritionist. Prescribing a therapeutic diet is, however, a complex technical act, the effectiveness of which is dependent on the accuracy of the indication and the patient’s adherence [33,34]. As such, dieticians intervene as experts in order to adjust the diet to the needs of the patients: they take part in menu creation and the adaptation from a standard diet to a therapeutic diet. Thus, through their interaction with both care and catering services, dieticians are guarantors of the quality of the food served, regardless of the disease [6,36,37]. Dieticians also contribute to the preparation of nutritional protocols, allowing all patients to benefit from a diet tailored to his or her needs upon admission, particularly in HEs with a limited number of dieticians [36].

The corollary of this recommendation is the following:**Recommendation 6**: It is recommended that all types of therapeutic diets be reassessed during hospitalization and upon discharge by the prescriber, and if necessary, by a dietician and/or a physician nutritionist.

Comment:

The implementation of a therapeutic diet in the case of nutritional disorder(s) is an act of care which should not derogate from the principle of re-assessment, as part of the quality process developed by the HAS [37].

The re-assessment of therapeutic diets during hospital stay has several objectives: to ensure the effectiveness and adherence of actions implemented, to adapt food provision to changes in clinical condition, and to propose, when necessary, oral, enteral or parenteral supplementation [6,11]. When the patient is discharged, this re-assessment is also useful for verifying the need for maintaining the therapeutic diet and its feasibility at home [13,34].

Given the often average and heterogeneous training level of physicians in the field of nutrition [34], the possible intervention by a physician-nutritionist and/or dietician for the re-assessment and adaptation of a therapeutic diet would appear advisable. The dietary care approach, involving the re-assessment during hospital stay or when patients are discharged, has been the subject of recommendations for good practices validated by the HAS [34]. Thus, in collaboration with numerous health professionals [6,21], the dietician plays a key role in this re-assessment [6,35]. Owing to the expertise provided by the qualitative and quantitative evaluation of patient nutrition [9,38,39], dieticians are hence qualified to control the quality of nutritional care and to propose adaptive measures [36].

Although this dietary re-assessment should ideally target all hospitalized patients with a therapeutic diet, the lack of personnel trained in nutrition and their distributive disparity according to health establishments are likely obstacles to its application. As a result, if the re-assessment by specialized personnel cannot be generalized within an HE, prioritized and targeted actions should, therefore, be carried out in the event of initiation of a therapeutic diet during an acute clinical situation, or when implementing a restricted or texture modified diet. Conversely, patients with stable chronic illnesses without change in clinical status or medical treatment likely do not require a prioritized and regular re-assessment of their therapeutic diet by nutrition specialists. In such cases, the re-assessment must, at minimum, be carried out by the prescribing physician.

**Recommendation 7**: It is recommended not to combine more than two restrictive therapeutic diets due to the risk of undernutrition.

Comment:

By “combine”, we imply the association (manual or computerized) of several types of prescribed diets.

Combining several levels of dietary restrictions renders the designing/preparation of a meal tray in the kitchen both difficult and random and exposes the patient to the possibility of reduced consumption and an increased risk of undernutrition [9,10,12]. It also impacts its acceptability from a visual and taste standpoint (for example, a prescribed texture-modified, no-residue, low-salt diet). It has a negative effect on the patient’s appetite. Many therapeutic prescriptions reduce the variety of foods consumed and expose the patent to a more monotonous nutritional diet. It also increases the risk of specific deficiencies (example: no-residue diet and vitamin C deficiency).

The prescription of a therapeutic diet is, therefore, a therapeutic medical act which encompasses the medico-legal responsibility of the prescriber [32] and incurs changes in the dietary status of the patient concerned. Dieticians are the health professionals qualified to define, assess and control the quality of food provision [36]. It is important to assess the benefit/risk ratio of restrictive diets, especially in the frailest patients, such as the elderly [40] and according to the recommendation of the French Council of Food and Nutrition (CNA, Conseil National de l’Alimentation et de la nutrition) [6]. The prioritization of needs should lead to limiting prescriptions to one or two therapeutic diets that are essential to the patient’s condition. Computerized meal ordering systems must limit the cumulative amount of therapeutic diets to a maximum of two possibilities. Prescriptions incorporating more than two restrictions must be the exception and should be managed solely by physicians or dieticians.

### 3.3. Adaptation of Food Provision

**Recommendation 8**: It is recommended that food provision can be partitioned by offering more than three meals a day.

Comment:

The current meal distribution system systematically includes three meals: breakfast, lunch and dinner. The possibility of offering snacks should constitute an integral part of the strategies to fight undernutrition. Health facilities must endeavor to provide food intake at times other than traditional mealtimes during the day, in the form of snacks. Health facilities should be able to fulfill the specific needs of certain patient populations (pregnant or lactating women [41], elderly people [27,42]), as well as implement strategies to fight undernutrition. Snacks should be given at coordinated time intervals with meals, but also to decrease the duration of night-time fasting [42].

**Recommendation 9**: It is recommended that the adaptation of food texture be established according to the International Dysphagia Diet Standardization Initiative (IDDSI) recommendations.

Comment:

Swallowing disorders affect nearly 8% of the world’s population and are particularly common in neurodegenerative diseases, stroke, cancers of the upper aerodigestive tract and in the elderly [43]. The appropriate adaptation of foods and beverages can reduce the complications of swallowing disorders, mainly undernutrition and lung infections. The adjustments must ensure a food intake and hydration that is safe, sufficient and pleasant for the patient. Experts from the European Society for Swallowing Disorders (ESSD) [44] conclude that a variety of choices must be available in order to adapt to the severity of the disorders, including modification of solid food texture (firmness, bite size, adhesion and consistency) and liquid thickness (transit speed) [44,45]. Aside from swallowing disorders, various conditions, such as locking of the jaw, dental extractions as well as certain post-ENT or post-bariatric surgery refeeding issues, may require the use of texture modifications.

Within the various care structures, there is a notable absence of common terminology to qualify the various textures proposed and in which the names, number of modification levels and characteristics are attributed empirically and in an extremely variable manner [45], leading to a very heterogeneous management of swallowing disorders. The thickening of liquids reduces the risk of pulmonary inhalation but increases the risk of post-swallow residues in the pharynx [46], and also necessitates the application of a liquid thickening grading system logged in the medical chart.

The IDDS initiative made it possible to achieve a universal, standardized terminology for adapting foods and beverages in the event of a swallowing disorder. The IDDSI recommendations were issued by a panel of international experts, including dieticians, after reviewing the international literature. These recommendations are based on a description of each adaptation level and specific measurement methods in order to foster their application in clinical practice. The recommendations are freely accessible on the https://iddsi.org/ website (accessed: 13 March 2021) and resulted in an eight-level diagram (from liquid to solid) (Figure 1). The characteristics, indications, measurement methods and examples of corresponding foods are detailed for each level. With the objective of rationalization, the health establishments must tend toward this classification by harmonizing the terms and foods proposed at each adaptation level according to the possibilities of each establishment and the type of patients received. One of the objectives of the IDDSI is to extend this harmonization to the products distributed by the agro-food industries, which will facilitate the work of catering professionals during the various purchasing procedures and the creation of menus.

Health establishments do not have to propose all variations of the textures in their menus but must identify their proposals according to the IDDSI classification. Indeed, the use of consistent terminology improves patient safety and communication between the various professionals as well as during transfer to another healthcare establishment invested in the nutritional well-being of the patient [44]. The use of an international standard tends to promote good practices and efficiency and to secure both professionals and patients alike. Given that this classification was published in 2016, it would seem legitimate to suggest establishing the texture adaptations according to IDDSI standards. It is recommended to refer to the IDDSI website, updated according to user feedback and translated into multiple languages.

### 3.4. Therapeutic Diets

**Recommendation 10**: During hospitalization, aside from a specialized indication, the prescription of a therapeutic diet aimed at weight loss is not recommended.

Comment:

In 2014, nearly half of the population in France was considered overweight. The prevalence of overall obesity was 15.8% in men and 15.6% in women, while the prevalence of abdominal obesity was 41.6% and 48.5% for men and women, respectively [47]. The increase in the prevalence of obesity affects all population age groups, including the elderly. As a result, the population of sick obese patients is also increasing in hospitals.

In hospitalized obese patients, the risk of undernutrition is high and often underestimated by healthcare professionals. Sarcopenic obesity is an emerging concern with very little literature regarding the subject. Its prevalence is estimated to be between 5 and 15% in the general population [48] and up to 22% in the non-hospitalized elderly population. In the hospital setting, there are currently no data relative to the prevalence of sarcopenic obesity.

In critically ill obese patients, it is recommended to provide 20–25 kcal/kg of ideal weight per day or <14 kcal/kg actual body weight per day as well as a protein intake of 1.2 to 2 g of protein/kg ideal weight per day [49]. There is no clear consensus for non-critically ill obese subjects, although recent literature data suggest that intake remain hypo-energetic with 20–25 kcal per kg of adjusted weight, albeit with a high protein component of 1 to 1.1 g of protein per kg of actual weight [50].

With regard to strategies related to sarcopenic obesity in the elderly, Batsis et al. [51] highlight the absence of evidence of specific diets. The authors emphasize the importance of combining dietary and lifestyle measures (limiting calorie restriction for the purpose of losing weight to 0.5 kg/week for a target of 8–10% loss in 6 months, followed by a stabilization phase, protein supplementation of 1–1.2 g/kg/day (25–30 g/d) along with calcium (food-based if feasible) and vitamin D supplementation, as well as physical activity to increase functional and cardiorespiratory capacities (aerobic and resistance exercises).

The 2010 recommendations for good clinical practice on preoperative nutrition [52] specify that obese patients are potentially malnourished patients and that an involuntary weight loss prior to surgery is an independent risk factor for complications. Thus, restrictive diets leading to a significant loss of lean mass are not recommended, particularly in patients with common obesity (BMI 30 to 40) or in elderly obese subjects. Voluntary preoperative weight loss is not recommended in the days and weeks before surgery. There is no evidence of the benefit of voluntary pre-operative weight loss regardless of the surgery. Finally, the recommendation specifies that if weight loss is necessary to facilitate the surgical procedure, a weight stabilization phase of at least 15 days is necessary before the intervention. It is also emphasized that it is probably not recommended to prescribe a low-calorie diet in an obese patient post-operatively.

Food intake should be considered an essential component of the treatment of hospitalized patients, even though many patients do not consume half of the meals served in the hospital, primarily due to anorexia linked to treatment and secondly to organizational or food provision reasons [12]. Based on calculations drawn from intake recommendations (excluding during resuscitation care), the Lausanne University Hospital team determined that both energy and protein requirements are not fulfilled when providing a standard diet of 1800 kcal and 60 g of protein per day for patients reaching the moderate obesity stage [53]. In cases of massive obesity and conditional to the total consumption of the meal tray, the energy deficit is approximately 500 kcal while the protein deficit is at least 55 g [53]. A study conducted by the same team showed that, on the one hand, the standard food provision was deemed not only insufficient but, in addition, under-consumption secondary to the disease worsened the protein-energy deficit; therefore, there was a risk of undernutrition and loss of lean body mass [54].

The prescription of a therapeutic diet aimed at weight loss is not recommended for patients with acute illness, under metabolic stress or surgical procedure.

On the contrary, however, the nutritional management of obese patients requires expertise and multidisciplinary management in order to assess the dietary intakes and individually adapt the latter to the calculated needs of the patients [53].

**Recommendation 11**: It is recommended to adapt protein intake according to Chronic Kidney Disease stage and nutritional status.

Comment:

The French Agency for Food, Environmental and Occupational Health & Safety (ANSES, Agence Nationale de Sécurité Sanitaire pour l’alimentation, l’environnement et du travail) recommend a protein intake of 0.83 g/kg/d of protein for healthy individuals.

In chronic kidney disease (CKD), the beneficial effects of controlling protein intake are manyfold: lowering of albuminuria and proteinuria, decrease in plasma and urinary urea, decrease in phosphatemia, reduction in insulin resistance and control of acidosis. These different factors not only act on renal health but also on cardiovascular, bone and neuromuscular health [55,56,57].

Protein intake should be adapted according to the stage of chronic kidney disease [58]:Normal renal function with increased risk of CKD (diabetes, hypertension, solitary kidney, etc.): <1 g of protein/kg/day or on adjusted weight if BMI > 30.Increase the proportion of vegetable proteins while maintaining a satisfactory animal protein/vegetable protein ratio.Mild to moderate CKD (stages II and III): 0.8 g protein/kg/dayConsider 0.6–0.8 g protein/kg/day if the glomerular filtration rate (GFR) <45 mL/min/1.73 m^2^ (stage IIIB) or if progression is rapid.Severe and end-stage CKD (stages IV and V): 0.6 g to 0.8 g/kg/d including 50% high biological value protein, or <0.6 g/kg/d with the addition of essential amino acids or keto-analogs.

Dialysis treatments (hemodialysis and peritoneal dialysis) lead to loss of proteins, which require readjustment of intakes greater than 1 g of protein/kg/d.

During these different periods, it is recommended to assess the nutritional status of chronic kidney disease patients by regular monitoring of weight, biological values (albumin), appetite and food consumption.

Any protein-energy undernutrition during renal failure and chronic dialysis requires an increase in protein intake from 1.2 to 1.4 g/kg/d, or even above 1.5 g in the case of hypercatabolism.

**Recommendation 12**: It is not recommended to prescribe a low-fat therapeutic diet < 35% of the total energy intake, with the exception of major primary hypertriglyceridemia and chylous effusions (chylothorax, chylous ascites and chyluria) in which a strict fat restriction is required (<30 g per day, excluding medium chain triglycerides (MCTs)).

Comment:

The above indication takes into account:The lack of consensus and scientific literature regarding the value of a low-fat diet in instances of hypertriglyceridemia, other than its primary form. There is no consensus as to the definition of major hypertriglyceridemia or of the threshold beyond which there is risk of acute pancreatitis.In the case of acute alcoholic pancreatitis, often associated with hypertriglyceridemia, the value of such a diet and the triglyceridemia threshold requiring a low-fat diet has not been established.Substitution with medium chain triglycerides is not addressed in this recommendation.The value of a low-fat diet (<35% of energy intake) is now questioned in its classic indications: prevention of obesity, prevention of cardiovascular disease [59]. The Mediterranean-type diet (40–45% fat, rich in monounsaturated fats and omega 3) appears more effective in terms of cardiovascular risk prevention, high blood pressure [60], prevention of type 2 diabetes [61], NASH (non-alcoholic steatohepatitis), etc.

Possible indications of low-fat diets, according to literature data:Hypertriglyceridemia (hyperTG):

Major hypertriglyceridemia occurring during an accumulation of chylomicrons is arbitrarily defined above a threshold of 10 to 15 mmol/L (8.8–13.4 g/L) [62]. Beyond this threshold, ingestion or infusion of triglycerides can increase hypertriglyceridemia and “clog” lipoprotein lipase (enzyme responsible for the breakdown of triglyceride-rich lipoproteins) by saturating its activity. This can lead to a rapid increase in triglyceridemia. The main objective is to avoid the occurrence of acute pancreatitis by containing triglyceridemia within a relative safe zone between 4 and 10 g/L [63], through a low-calorie and/or low-carbohydrate and/or alcohol-restricted diet. The value of a low-fat diet is not established in this indication with the exception of the primitive forms in which an extremely low-fat diet (limiting fat intake to 30 g or even 10 g per day) is required. However, the risk of pancreatitis increases when TG levels exceed 10 to 15 g/L [64,65] although concentrations between 2 and 10 g/L can be observed at the time of diagnosis of acute pancreatitis regardless of etiology [66,67]. The risk of acute pancreatitis in patients with a serum TG at 10 g/L is 5% and 10–20% at 20 g/L [65]. According to certain authors [68], a hyperTG > 10 g/L, therefore, requires a restriction of dietary fat to between 10% and 15% of calorie intake, with a reduction in saturated and unsaturated fat ranging from 10 to 25 g.

Acute pancreatitis:

Hypertriglyceridemia is one of the causes of acute pancreatitis. Aside from hypertriglyceridemia linked to type 1 (hyperchylomicronemia) or type 5 dyslipidemia, which can be responsible for acute pancreatitis, hypertriglyceridemia is generally associated with alcohol consumption, even in small quantities (subject identified as “sensitive” to alcohol). According to certain authors, care should be taken to monitor the concentration of plasma triglycerides [66].

Exocrine pancreatic insufficiency: chronic pancreatitis:

There is no indication for restricting the proportion of dietary fat. Increasing pancreatic enzyme replacement therapy reduces steatorrhea. Addition of a proton pump inhibitor can increase therapeutic effectiveness. On the other hand, the patient must be persuaded not to limit his or her diet. The measurement of steatorrhea under treatment makes it possible to assess the effectiveness of treatment as well as calorie loss (9 kcal/g of fat). In the case of major malabsorption (residual steatorrhea > 40 g/d) and undernutrition, medium chain triglycerides (20 to 30 g) can be introduced, although such intervention is exceptional. Eighty percent of patients can, therefore, continue a normal diet if combined with pancreatic extracts [68].

Following supramesocolic surgery (cephalic duodenopancreatectomy, gastrectomy and superior polar esogastrectomy):

In contrast, an increase in calorie intake, in all forms, is strongly encouraged. Treatment with pancreatic enzymes should help control steatorrhea. Treatment should be optimized by controlling the level of steatorrhea under treatment.

Chylothorax:

A low-fat diet (<10 g/d) is generally recommended, and typically associated with other treatments (pleurodesis). Although no case/control study was able to demonstrate its value, descriptive data in one study showed that this diet was associated with a resolution of chylothorax in two-thirds of the cases with a median follow-up of 10 days (5–27 d). The combination of a low-fat diet and pleurodesis resolved the chylothorax in more than 80% of cases not responding to diet alone. In this latter study, the indication retained for pleurodesis was the presence of >500 mL of chylous fluid during the first 24 h after initiation of the low-fat diet [69].

Chyluria:

A low-fat diet, combined with or without somatostatin, allows reducing lymphatic leakage. It is sometimes necessary to combine the former with etiological treatments (antiparasitic, surgical treatment, etc.) [70].

Chylous ascites:

Abundant chylous ascites, greater than 1 L in volume, require a maximum reduction in lymphatic flow. Lymphatic flow increases from 1 mL/min to 200 mL/min after a fat-containing meal. Normal food maintains lymphatic flow as a result of long-chain fatty acids entering the lymphatic pathway after their enterocytic absorption. A strict fat-free oral diet leads to a significant calorie restriction, since there are very few foods that do not contain long-chain fatty acids (fruits and vegetables) and the foreseeable duration of such management should not exceed 7–10 days and, as such, requires the indication of parenteral nutrition. In the initial phase, in the presence of large ascites, exclusive parenteral nutrition is indicated in order to significantly reduce lymphatic flow. Enteral nutrition is indicated only in cases of small-volume chylous ascites. In France, among the family of commercially available solutes intended for enteral nutrition, semi-elemental solutes are those which contain the highest concentration of medium chain triglycerides (MCT) compared to standard polymeric solutes (45–70% versus 15–25%) [71,72,73,74,75].

Bile acid malabsorption:

A randomized study has shown that a low-fat diet can help reduce symptoms related to bile acid malabsorption (biliary diarrhea). However, the impact of such a diet on the patients’ nutritional status was not assessed [76].

Obesity and cardiovascular risk:

Hospitalization in acute care is not the time to initiate a low-calorie or low-fat diet.

Results of several meta-analyses of randomized studies comparing low-carbohydrate or low-fat diets (<30% of energy intake), combined or not, with a reduction in energy intake in strictly adherent populations have shown that each diet was associated with a significant weight loss and reduced predicted risk of cardiovascular events. However, the low-carbohydrate diet was associated with greater weight loss and greater improvement in cardiovascular risk [77,78,79,80].

A low-fat diet is associated with a decrease in LDL-cholesterol, but an increase in triglyceridemia and a reduction in HDL-cholesterol [81]. This reduction in HDL-cholesterol can be limited by adding monounsaturated fatty acids (e.g., olive oil) [82].

In the case of diabetes, a low-fat diet does not appear to alter blood sugar balance in the medium term, compared to a low-fat diet [83,84].

Cancer:

Increased consumption of fruits, vegetables and grains is associated with a decrease in breast cancer mortality in a large cohort study. A fruit- and vegetable-enriched diet leads to a decrease in fat consumption [85].

NASH (non-alcoholic steatohepatitis):

Calorie restriction leads to an improvement in fatty liver and NASH, irrespective of intake distribution. The most effective approach appears to be the implementation of a Mediterranean diet characterized by a decrease in carbohydrate intake, in particular sugars and refined carbohydrates (40% of calories compared to 50–60% in a low-fat diet), and by an increase in monounsaturated and omega-3 fatty acid intake (40% of calories in the form of fat compared to <30% in a typical low-fat diet) [86,87,88,89]. An isocaloric diet rich in monounsaturated fatty acids leads to a reduction in fatty liver disease [90].

Nevertheless, a high-fat diet exacerbates fatty liver disease [91,92], while a low-lipid isocaloric diet could be of interest provided that it is not compensated by an increase in the consumption of fructose or carbohydrates with a high glycemic index [92].

**Recommendation 13**: The standard diet is suitable for diabetic patients without exclusion of foods and desserts containing sucrose.

Comment:

For nutrition recommendations, France refers on the one hand to French public health data, and on the other, to the recommendations of the ADA (American Diabetes Association) [93], which remain in agreement in both North America and Europe.

The recommendations for macronutrient intake in individuals with diabetes are the same as for the general population, in the absence of diabetes complications.

Sugar and sweet products can be consumed in the same quantity as the recommended intakes for the general population.

As advocated by ANSES, it is more a question of “controlling” the consumption of sugar as opposed to excluding the latter. The lowest contribution identified in the literature from which an alteration in risk markers is observed has been considered. The minimum consumption for which a significant increase in blood triglyceride concentrations was observed is 50 g of fructose per day, i.e., 100 g of sucrose (ANSES, 2016) [94]. Thus, an upper limit of 100 g/d has been set for the total consumption of sugars, excluding lactose and galactose [94]. This particular point is not required in hospital catering. For an optimization of carbohydrate distribution, their consumption should be considered in equivalence to that of other carbohydrate foods [95,96], with the exclusion of sugary drinks whose consumption should be avoided outside of adapted situations.

With the exception of the consumption of sugar-sweetened beverages, there are, therefore, no recommendations which exclude sucrose from the diet of individuals with diabetes [97,98], within the framework of compliance with public health policies. From a scientific standpoint, there are no arguments to contradict these recommendations. This was aptly demonstrated over 35 years ago [99].

Note: In gestational diabetes, there is also no systematic exclusion of sucrose. It is rather the choice of foods according to their glycemic index that is recommended [100].

**Recommendation 14**: It is recommended that the standard diet provide a regular carbohydrate content for each meal.

Comment:

The standard diet is appropriate for patients with diabetes. For those who do not adjust their insulin dose according to carbohydrate intake (functional insulin therapy), the amount of consumed carbohydrates must be regular and at fixed times. This favors glycemic balance and minimizes glycemic excursions including the risk of hypoglycemia [93].

The designing of menus with regular carbohydrate content allows standard food to be offered to diabetic patients. Regular intake includes the entire meal, not just starchy foods. Indeed, the amount of carbohydrates in the meal should be calculated and planned taking into account all of the components of a meal. This approach will allow more flexibility in menu creation (e.g., starchy products as appetizer, main dish, dessert or bread).

**Recommendation 15**: It is recommended that the amount of carbohydrates be known and accessible for each dish served.

Comment:

Individuals under functional insulin therapy must be able to estimate the amount of carbohydrates consumed, and possibly fats and proteins, in order to define the adjusted mealtime insulin dose for better glycemic control [93].

Recommendation 4 on accessible nutritional values will facilitate the achievement of this recommendation.

Insofar as recommendations 2 (standard diet meeting the recommendations for prevention and health promotion) and 4 (known nutritional values of provided foods) are respected, the standard (so-called normal) diet may be offered to individuals hospitalized with diabetes. The meals offered should meet public health criteria with regard to diabetic diet provision on the one hand, and on the other, the amount of carbohydrates per dish should be given.

**Recommendation 16**: If a low-salt therapeutic diet is indicated, it is recommended not to restrict salt intake (NaCl) from the diet to less than 5 g/day (or approx. 2 g of sodium/d), except in severe acute decompensation and for a very short duration.

Comment:

(1) In heart failure:

There is insufficient scientific evidence to support sodium restriction [101]. Several studies are ongoing to assess the effects of a restricted salt (NaCl) consumption in the setting of heart failure (GOURMET-HF [102], SODIUM-HF [103], ProhibitSodium [104]) and should enable the providing of answers in the coming years.

However, the majority of learned societies recommend a low-sodium (Na) diet, particularly in the case of acute cardiac decompensation [105].

The European consensus recommends a moderately restricted sodium (Na) diet, without specifying the threshold [106]. Finally, the American [107], New Zealand, Canadian and Australian [108,109] recommendations advocate a sodium (Na) consumption of less than 2 g per day, or 5.1 g of salt (NaCl). All of these recommendations are based on expert agreement.

A diet under 5 g of salt (NaCl) or ~2 g of sodium (Na) per day exposes the patient to a major risk of undernutrition by a reduction in food intake [12] due to the lack of flavor of the dishes: there is no indication in cardiology.

(2) In the setting of ascitic cirrhosis, the European Association for the Study of Liver recommends a restricted diet of between 4.6 and 6 g of salt (NaCl) or ~1.8 to 2.4 g of sodium (Na) per day. These recommendations are based on two randomized trials [110]. It is specified that a more restricted salt diet is often associated with a reduction in oral intake and a risk of undernutrition or worsening thereof.

(3) In nephrology, a sodium (Na) consumption of less than 4 g per day (or <10 g per day of NaCl salt) is recommended. In the case of volume overload (edema), sodium (Na) consumption should be less than 3 g per day (or 7.6 g of NaCl salt) [58].

**Recommendation 17**: The diet designated as “acid-free and/or spice-free” has no indication (except oral or digestive hypersensitivity or food allergy).

Comment:

The terminology of “acid-free and spice-free diet” is that reported by health establishments offering this type of diet. The purpose of the present recommendation is to eliminate this type of diet which has no scientific basis aside from oral sensitivity, particularly for certain spices, which may be related to mucosal lesions.

### 3.5. Glossary

Spices: A spice is an aromatic or pungent vegetation or mineral product. A spice-free diet, however, essentially targets hot spices, comprising piperine or capsaicin (pepper, chili, etc.).

The digestive effects of the spices described in the literature are rather positive, although the level of evidence is low, with the exception of a few indications.

Spices may improve digestion by stimulating certain digestive enzymes. They can potentially increase satiety [111], particularly capsaicin. A spicy, more satietogenic diet has a limiting effect on food intake and may facilitate weight gain management [112]. Certain studies have shown that pungent spices may increase plasma concentrations of zinc, iron and calcium (piperine, capsaicin and ginger) [113] and beta-carotene [114], which may explain the antioxidant effect reported in certain studies [115,116]. The negative association between spicy food consumption and LDL-cholesterol has led to the suspicion of a cholesterol-lowering effect [117], which has been corroborated in intervention studies, although with a low level of evidence and low statistical power, particularly with ginger and turmeric [118]. The consumption of spices may reduce salt intake, which is implicated in hypertension [119] and gastric carcinogenesis. An antineoplastic effect of capsaicin has been reported in several laboratory studies, prompting the testing of analogs [120], and anticancer effects have also been found for curcumin [121]. Among potential negative effects, certain spices may inhibit a number of liver enzymes involved in the metabolism of drugs, cytochromes P450 [122], in particular capsaicin (CYP2C9), piperine (CYP3A4) and curcumin [123]. Hot spices can increase digestive permeability, which may explain an increase in allergen sensitivity in the event of food allergy [124], and may also be involved, according to some authors, in the modulation of certain autoimmune diseases [125].

### 3.6. Beliefs and Misconceptions

In irritable bowel syndrome: no study has established a link between spicy food and symptoms.In gastroesophageal reflux disease (GERD): spices may induce burns, but not GERD. Spices can indeed trigger pain when there are existing lesions, such as esophagitis or gastric ulceration. Spices are not involved in the pathophysiology of lesions, but may reveal these lesions by their hyperemic effects [126].Spices have no involvement in the pathophysiology of ulcers. Some studies have shown that spices may increase mucus secretion and have a protective role. Capsaicin has been reported to inhibit acid secretion, stimulate mucus secretion and gastric mucosal blood flow, thereby helping in the prevention and healing of gastric ulcers [127]. The metabolic pathway involves prostaglandin E_2_ and prostacyclins in conjunction with EP1 and the IP receptor [128,129]. Spices (pepper, chili) thus appear to have a protective effect on the gastric mucosa.

### 3.7. Oral Mucositis

In the case of oral mucositis, a study has shown that the symptoms are significantly correlated with the consumption of spicy and/or hot-temperature foods [130].

### 3.8. Regarding the Acidity of Food

There are a number of misconceptions and beliefs regarding the acidity of food. For example, honey has a more acidic pH than tomatoes (3.7–4.2 vs. 4.3–4.9) and grenadine syrup has a lower pH (2.31) than vinegar (2.40–3.40), lemon juice (2.4–2.6) or cola (2.5). No food reaches a gastric pH equal to 2.

No study has established a link between the pH of food and digestive lesions or diseases. As is the case with spices, certain beverages or foods can trigger pain in the event of a preexisting lesion (esophagitis for example), although the link with the pH of the food is not established. Most acidic foods are perfectly tolerated, even with mouth lesions. Conversely, combining simple sugar with an acidic pH increases the risk of dental erosion [131].

**Recommendation 18**: It is recommended not to exclude pulp-free fruit juices, potatoes, white bread, milk and dairy products from a “strict low-fiber” diet (10–14 g fiber/d, commonly called low-residue or no-residue diet).

Comment:

The designations pertaining to the types of food targeting residues and fibers are subject to semantic heterogeneity from one health establishment to another, both in France and worldwide. As a result, it is possible to find designations as varied as “strict no residue”, “no residue”, “low-residue”, “no fiber”, “low-fiber”, “digestive restriction”, “light”, etc. In addition, the same designation can correspond to different levels of restriction. In practice, in clinical studies, the terms “low-fiber” and “low-residue” have the same meaning [132]. This is, therefore, a source of confusion for both the prescriber and the patient.

Regardless of the designation, the goal of this type of diet is to limit the volume and number of stools and gas.

### 3.9. Definition and Semantic Choices

In accordance with Regulation (EU) No. 1169/2011 [133], dietary fiber means:

Carbohydrate polymers with three or more monomeric units, which are neither digested nor absorbed in the human small intestine and belong one of the following categories:
Edible carbohydrate polymers, naturally occurring in the food as consumed;Edible carbohydrate polymers which have been obtained from raw food material by physical, enzymatic or chemical means and have a beneficial physiological effect demonstrated by generally accepted scientific evidence;Edible synthetic carbohydrate polymers which have a beneficial physiological effect demonstrated by generally accepted scientific evidence.

While it is necessary to be aware of the regulatory definition of dietary fiber, which highlights the health effect of fibers, it is also imperative, in therapeutic practice, to keep in mind that there are other compounds comparable to fibers which do not meet this definition but which, similarly to fibers, are not digested in the small intestine and can have digestive effects, including lignin, polydextroses or even, under certain conditions, lactose or fructose, etc. In its scientific opinion of 2010, l’EFSA (the European Food Safety Agency) gives a broader definition which includes this type of compound [134,135,136].

A residue is defined as the food fraction, derived from fibers or assimilated to fibers, which is not degraded in the small intestine under physiological conditions, and which increases the volume of stool and or gas [132,137].

The following choices are proposed and justified:The elimination of the designation “no residue”: this designation should no longer be used because all foods generate residues.The term “low-fiber” replaces that of “low-residue”. In some countries such as the United States, the learned societies in charge of reflecting on these themes have proposed since 2011 to prefer the term “low-fiber” to that of “low-residue”; indeed, unlike fiber, there is no consensus scientific methodology to accurately assess the residue content of a meal [132,138]. Similarly to this search for harmonization across the Atlantic, and anxious to base itself on a quantifiable scientific definition, it is proposed to no longer use the designation “low-residue” diet.To define fiber intake thresholds according to the level of restriction:
“strict low-fiber” corresponds to intakes of 10 to 14 g/d of fiber;“low-fiber” corresponds to intakes of 15 to 20 g/d of fiber;A diet consisting of less than 10 g/d of fiber has no indication since it has not been proven to have a therapeutic or diagnostic benefit additional to that of a fiber intake between 10 and 14 g/d [132,139].As a reminder, most countries recommend that healthy adults, as part of a balanced diet, consume 25 to 35 g/d of dietary fiber daily [134]. In France, ANSES recommends a consumption of 30 g/d while the average consumption among French adults according to the INCA3 survey is 20 g/d [140,141]. The “strict low-fiber” diet (10 to 14 g/d):

This diet must meet daily energy and macronutrient needs.

Compared to the aforementioned definitions, the classic exclusion of white bread, potatoes, pulp-free fruit juices, milk and dairy products in the “strict low fiber” diet is currently not based on any rational argument [142,143,144].

Fibers and, in particular, those in fruits and vegetables favor GI transit [145]. Due to their effect on the volume of stools and gases and on transit, the following are excluded: whole grains, and vegetables and fruits including apple and prune juice (other pulp-free juices can be proposed). In light of several old studies [143], there is, however, no justification to prefer stale bread, toast or rusks, to fresh bread. Potatoes can be introduced in any form.

Lactose can be poorly absorbed under certain conditions and induce digestive disorders. Intolerant individuals, however, represent only a small proportion of the lactose-malabsorption population. Lactose intolerance, whether primary or secondary, is managed by adapting the consumption of the products to the patient’s tolerance without eliminating them [146] (see recommendation 21). There is, therefore, no justification for systematically excluding milk and dairy products from the “strict low-fiber” diet.

It is important to note that the “strict low-fiber” diet remains nonetheless restrictive and monotonous and can induce a nutritional risk. It should be prescribed for a limited period and reassessed regularly.

**Recommendation 19**: It is recommended to reserve the “strict low-fiber” diet (10 to 14 g/d of fiber) for therapeutic purposes in symptomatic intestinal strictures; for diagnostic purposes in certain GI explorations (colonoscopy, CT colonography, MR enterography (MRE), etc.); or for symptomatic purposes.

Comment:

The “strict low fiber” diet provides 10 to 14 g/d of fiber, as defined in recommendation 18.

The indication of a “strict low-fiber” diet for therapeutic purposes is not based on a strong scientific rationale outside the field of symptomatic intestinal strictures (sub-occlusion) and lower digestive fistulas (small intestine and colon) [146].

It is recommended in colonoscopies in addition to or as a replacement for purging when administration is impossible and, in such cases, constitutes an indication of diagnostic value [147]. There is lack of consensus regarding the prescription period, which can range from 1 to 3 days, or even be limited to a meal. In the American recommendations, the duration used for such a diet in addition to the purge is equivalent to 1 day, the day before exploration. This therapeutic diet can be prescribed for other explorations, in particular for imaging, to improve the quality of images by reducing the production of intestinal gas. There is no rationale for higher restriction levels, namely a fiber intake of less than 10 g/day (see recommendation 18).

Fibers participate in nutritional balance; furthermore, it is advisable not to unduly restrict fibers, including in gastrointestinal diseases. The prescription of a “strict low-fiber” diet for therapeutic purposes is not recommended in colonic diverticulitis [148,149], in irritable bowel syndrome, or in IBD [150]. In case of type III chronic intestinal failure and irrespective of the type of short bowel, it is not recommended to proscribe fibers, including in type 1 short bowel (jejunostomy), since fibers slow down gastric emptying and contribute to energy recovery in type 2 and 3 short bowel via the intestinal flora of the colon [151,152].

While there is no indication to prescribe a strict low-fiber diet for therapeutic purposes in these diseases, there may, however, be indications to prescribe this type of diet on a case-by-case basis, and transiently, for symptomatic relief, in order to limit intestinal distension, diarrhea, excessive flatulence or digestive pain. This can be the case in acute or chronic and active (or flare-up) enteropathies such as small bowel radiation enteritis or inflammatory bowel disease.

This diet modality is aimed at improving symptoms experienced by the patient and, consequently, to improve quality of life. It must be personalized, and its relevance reassessed after treatment of the underlying disease which led to its indication. The duration should be as short as possible in order to limit nutritional risk.

In the case of colonic diverticulitis, low levels of evidence do not make it possible to substantively define an optimal level of fiber intake. The HAS recommends not restricting fiber intake in such patients [148]; however, pending recommendations based on stronger evidence, some authors suggest limiting the intake of fiber in case of pain [153,154], while other authors propose leaving the patient free to consume whatever amount he or she deems able to tolerate during the acute period [155].

**Recommendation 20**: Per medical prescription, a low-fiber diet (15–20 g fiber/day) may be indicated during hospital stay, in terms of digestive symptoms.

Comment:

Although there are no or very few studies with strong scientific evidence which recommend a low-fiber diet providing between 15 and 20 g fiber/day, it can help improve quality of life in some patients. This diet aims to limit the volume and number of stools and gases to a lesser degree than the strict low-fiber diet. It is essentially based on professional expertise.

It is an intermediate level between the “strict low-fiber” diet and the standard diet (fiber intake between 25 and 30 g/d). It enables the limiting of nutritional risks linked to restrictions that are too high relative to the expected benefits. These have been widely described in the literature with anorectic effects, imbalances, insufficient energy intake, etc. The strict low-fiber diet, for example, is associated with monotony and a deficiency in vitamin C intake [12].

Conversely to the strict low-fiber diet, in addition to pulp-free fruit juices, the low-fiber diet incorporates vegetables and fruits whose fiber content is less than 3 g/100 g (while continuing to exclude whole grains, legumes and dried fruits). In Europe, a food can be considered a source of fiber if it contains at least 3 g of fiber/100 g or 1.5 g of fiber/100 kcal. It is considered high in fiber if it contains 6 g of fiber/100 g or 3 g of fiber/100 kcal [156]. Food fiber contents are available in the CIQUAL table [157].

Seasoning fat is not limited. This diet, therefore, allows a better diversity of proposed dishes.

With regard to current compositions [157], the total fiber intake of this low-fiber diet is between 15 and 20 g of fiber/day. This dietary mode is similar to the most widespread diet in the French population [141], even though ANSES recommends a consumption of 30 g/day for healthy adults [140].

The low-fiber diet can be indicated for therapeutic purposes for diseases requiring a moderate level of fiber restriction, such as mild or non-symptomatic intestinal stricture or gastroparesis [158,159,160].

It can also be indicated for symptomatic relief when expanding a strict low-fiber diet, and also in GI replenishment (knowing that there is no indication to prescribe a strict low-fiber diet in this setting).

This type of diet aims to preserve digestive comfort while limiting restrictions, the secondary benefit, therefore, being to improve intakes.

This diet must be prescribed for a limited period, be adapted to individual tolerance and be reassessed.

**Recommendation 21**: It is recommended not to exclude all dairy products in the context of lactose intolerance.

Comment:

In order to be absorbed, lactose must first be hydrolyzed by lactase. While lactase deficiency is common, we only speak of intolerance when the malabsorbed lactose is responsible for digestive disorders (by osmotic effect in the small intestine and or by fermentation in the colon). On the other hand, this component of malabsorbed lactose leads to colonic adaption by the microbiota, which could prove favorable [161,162].

Lactase deficiency, common in adults, does not mean that you are lactose intolerant. Intolerance represents only a small proportion of maldigesters, the vast majority of lactose–malabsorbing individuals being asymptomatic [163].

The Lactose Breath Test (25 g) remains the benchmark method of diagnosis. It is used to assess malabsorbed lactose. It must be accompanied by assessment of symptoms (for 8 h) which defines the level of tolerance [164,165,166].

Lactose intolerance is not an allergy. When the diagnosis of lactose intolerance is made, management does not consist in suppressing, but in adapting the consumption of milk and dairy products according to the patient’s tolerance, the objective being to eliminate annoying digestive disorders.

While tolerance thresholds vary according to the individual, studies have shown that maldigesters can tolerate up to 12 g of lactose (i.e., a glass of milk) when it is consumed alone and on an empty stomach, and up to 20 g when ingested with other foods [167].

Even with an equal lactose content, foods are not all equal in terms of tolerance due to their composition (lactose load, fat content, etc.), texture and their combination, or not, with other foods. Anything that slows down gastric emptying can improve lactose tolerance.

Milk as a beverage is the least tolerated form (liquid), especially if it is skimmed and taken on an empty stomach.

Whether it is a primary intolerance or disease-related intolerance, it is these concepts that should be relied upon to improve tolerance. It is not necessary to eliminate lactose from the diet [143,151,168].

In all instances, there is no justification for eliminating yogurts and cheeses, or foods that are sources of low lactose content [169].

Milk and dairy products are a source of calcium and protein. Their removal constitutes a risk to bone health; hence, there should be no unnecessary restriction, especially since the adaptation of the colonic microflora (colonic acidity) to digest lactose is one of the mechanisms of lactose tolerance [163,170,171,172].

**Recommendation 22**: Aside from medically diagnosed celiac disease, a gluten-free diet is not recommended.

Comment:

Celiac disease is an autoimmune-mediated enteropathy. It is also called gluten intolerance, which can cause confusion regarding the nature of the condition.

In adults, the diagnosis of celiac disease is based on the detection of specific serum antibodies, confirmed by a duodenal biopsy [173].

Permanently eliminating gluten is the only treatment for celiac disease.

Herpetiformis dermatitis is a skin expression of celiac disease, which also requires the monitoring of a gluten-free diet [174,175,176].

The definition of non-celiac gluten sensitivity (NCGS) initially retained by the experts is: “syndrome characterized by intestinal and extra-intestinal symptoms related to the ingestion of gluten-containing food, in subjects that are not affected by either celiac disease or wheat allergy” [177].

In light of the current literature, SGNC is a disease that is still poorly understood, the mechanism of which is likely multifactorial. Other components of wheat, rye and barley are likely to be responsible for the observed disorders (ATI-amylase trypsin inhibitor-, WGA-wheat germ agglutinins-, FODMAPS-fermentable oligo-, di- and monosaccharides and polyols-, etc.). For this reason, the name SGNC is still under discussion, with some authors proposing the term “wheat sensitivity”.

The responsibility of fructans (FODMAPS) is often mentioned, although there is ostensibly no point in preferring gluten-free cereals to traditional cereals with regard to fructan content [178,179].

In patients with irritable bowel syndrome, the current diagnosis of SGNC is a diagnosis of exclusion which, according to the recommendations of the experts, is achieved in two stages, after ensuring that the patient has a balanced diet: assessment of the effects of removing gluten (at least 6 weeks); impact of its reintroduction (at least 4 weeks) [178,180].

Once diagnosis has been established, the duration of exclusion and reintroduction are not defined. Insofar as the treatment aims only to limit the disorders, it would appear logical to adapt the consumption to the tolerance of the patient, and not by imposing a strict diet for life.

While the link between gluten and celiac disease is well established, the responsibility of gluten in SGNC remains to be demonstrated [181]. It is, therefore, not possible to date to state that a gluten-free diet is indicated in SGNC.

Wheat protein allergy involves complex phenomena. Allergic reactions to wheat proteins are caused by various types of exposures—ingestion, inhalation, contact—and can also appear following physical exertion. Most often, this allergy is transient and can disappear with growth. Gluten proteins are not the only elements likely to be responsible for wheat allergy. A gluten-free diet is, therefore, not indicated in the context of wheat protein allergy, since the treatment is not limited to solely gluten restriction [182,183,184].

In all instances, whether it is the application of the diagnostic protocol, or the adaptation of the diet once the diagnosis has been established, dietary management is necessary to support the patient and limit the risks of nutritional imbalance.

Not only is the removal of gluten not indicated outside the diagnosis of celiac disease, this removal can also be harmful by incurring an imbalance in nutritional intake [185,186].

When there is no medical indication for a gluten-free diet, it should nevertheless be possible to offer suitable and balanced food provision to patients who have deleted gluten out of habit, conviction or tolerance issues. It is indeed advisable not to induce a nutritional risk by proposing a meal to the patient which he or she will not consume.

**Recommendation 23**: It is recommended to propose, in terms of food provision, an energy- and/or protein-enriched diet in order to meet the requirements of prevention and management of undernutrition.

Comment:

Proposing an energy- or protein-enriched diet encompasses 30 to 50% of undernourished hospitalized patients or those at risk of undernutrition. Together with the standard diet, proposed by recommendation 2 (meeting the recommendations for promotion and prevention of health), this enrichment is aimed at prioritizing the prevention and management of undernutrition by making it possible to meet the needs of patients who have trouble feeding themselves properly (lack of appetite, staggered or skipped meals due to exams or fasting, etc.) through enrichment strategies.

Several strategies are possible to increase the energy and/or protein density of meals and allow the increasing of both energy and protein intakes, particularly in the elderly [187]. These strategies aim to offer: enriched meals through foods with high nutritional density (milk powder, dairy products, cheeses, fats and sugars), added to various salty or sweet preparations (soups, sauces, dessert creams, cakes and breads), and to adjust the size of the portions served. A systematic review by Mills et al. (2018) [188] showed the beneficial effects of meal fortification strategies and the provision of snacks in ten studies totaling 546 patients. Intake was significantly increased compared to current care by 250 to 698 kcal per day and 12–16 g of protein per day, according to the studies. The cost-effectiveness was favorable (measured in two studies), and patients expressed good acceptability of the enriched products while appreciating the taste (in four studies).

The goal is to offer higher energy- and/or nutrient-dense meals, without increasing their volume.

The other possibility is to reduce the volume of meals for patients that are small eaters, in order to favor consumption. This in turn necessitates enrichment and supplementation strategies in the form of snacks, in order to reach the daily nutritional target [189].

This requires flexibility on the part of both the healthcare staff and catering services [6], and an adaptation to the patients’ personal food/mealtime experience during their hospitalization [190].

## 4. Conclusions

All efforts should be made to create meals that follow these recommendations while promoting the taste quality of the dishes and their presentation such that the patient rediscovers the pleasure of eating in the hospital.

To promote the implementation of the guidelines, the AFDN and the SFNCM have just made available to health care institutions a grid for assessing current practices and deviations from the recommendations. This will allow the establishment of an implementation plan with priorities. At the same time, practical information sheets are being developed to provide practical assistance to hospitals. They are based on targeted arguments to convince both healthcare and catering professionals and are also based on the successful experiences of innovative establishments.

Furthermore, we have obtained the labeling of the guidelines by learned societies in the fields of diabetes, nephrology and gastroenterology. This is a cornerstone in ensuring the acceptance of the recommendations by healthcare professionals and patients alike.

## Figures and Tables

**Figure 1 nutrients-13-02434-f001:**
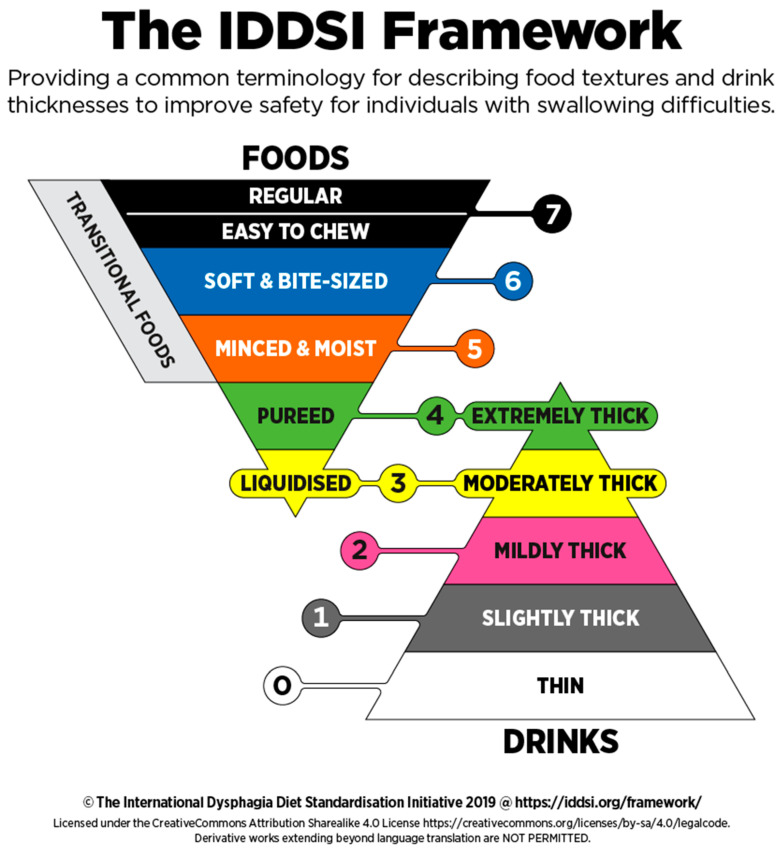
Diagram of the 8 levels of texture (from liquid to solid) according to the International Dysphagia Diet Standardisation Initiative (IDDSI).

## Data Availability

Not applicable.

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
