# Peer review of "Guidelines on Standard and Therapeutic Diets for Adults in Hospitals by the French Association of Nutritionist Dieticians (AFDN) and the French Speaking Society of Clinical Nutrition and Metabolism (SFNCM)"

_nutrients, 2021, doi:10.3390/nu13072434_

Round 1
Reviewer 1 Report
It appears to be useful. While the recommendations appear sound, will this make patients happier when it is put before them ? A recommendation in the conclusion that suggests creating meals that follow nutrition recommendations work any more tasty.
Author Response
Reviewer 1:
English language and style are fine/minor spell check required
Response: The manuscript has been revised linguistically by a native English speaker
It appears to be useful. While the recommendations appear sound, will this make patients happier when it is put before them ? A recommendation in the conclusion that suggests creating meals that follow nutrition recommendations work any more tasty.
Response:
As suggested, we have added this sentence at the hand of the abstract (line 53-55) and in the conclusion section (line 1115-1118):
“All efforts should be made to create meals that follow these recommendations while promoting the taste quality of the dishes and their presentation such that the patient rediscovers the pleasure of eating in the hospital”

Reviewer 2 Report
#1 Although some potential users were stated in lines 98-99, the target users of this guideline should be detailed: who will or will not be considered to use this guideline?
#2 Recommendation 10 ought to be along with commentary. In the Comment section, the authors stated “It is therefore not recommended to implement a restrictive therapeutic diet during hospitalization in the case of acute illness, during metabolic stress or surgery”. Would this Recommendation statement be more appropriate as “The prescription of a therapeutic diet aimed at weight loss is not recommended for the patients with acute illness, under metabolic stress or surgery” or like that?
#3 Please state the plan for how the investigators plan to reflect these recommendations in clinical practice.
Author Response
Reviewer 2:
#1 Although some potential users were stated in lines 98-99, the target users of this guideline should be detailed: who will or will not be considered to use this guideline?
Response: This paragraph has also been revised as suggested (line 100-105).
These recommendations are intended for the personnel involved at every level of the food chain, including people involved in patient care: catering industry overseers (catering engineer or kitchen manager), catering assistants, care workers; nursing staff supervisors: managers, senior health managers and care coordinators; dieticians, nurses, caregivers, hospital services agents, physicians, upper management overseers such as hospital heads, purchasing or economic services directors / logistics directors, and institutional stakeholders.
#2 Recommendation 10 ought to be along with commentary. In the Comment section, the authors stated “It is therefore not recommended to implement a restrictive therapeutic diet during hospitalization in the case of acute illness, during metabolic stress or surgery”. Would this Recommendation statement be more appropriate as “The prescription of a therapeutic diet aimed at weight loss is not recommended for the patients with acute illness, under metabolic stress or surgery” or like that?
Response: We agree that this rephrased sentence is more accurate.
Line 541-542: The prescription of a therapeutic diet aimed at weight loss is not recommended for patients with acute illness, under metabolic stress or surgical procedure.
#3 Please state the plan for how the investigators plan to reflect these recommendations in clinical practice.
Response: As suggested, we have added this perspective in a conclusion section, added at the end of the manuscript (line 1115-1125).
- Conclusion
All efforts should be made to create meals that follow these recommendations while promoting the taste quality of the dishes and their presentation such that the patient rediscovers the pleasure of eating in the hospital.
To promote the implementation of the guidelines, the AFDN and the SFNCM have just made available to health care institutions a grid for assessing current practices and deviations from the recommendations. This will allow the establishment of an implementation plan with priorities. At the same time, practical information sheets are being developed to provide practical assistance to hospitals. They are based on targeted arguments to convince both healthcare and catering professionals and are also based on the successful experiences of innovative establishments.
Furthermore, we have obtained the labeling of the guidelines by other learned societies in diabetes, nephrology, and gastroenterology. This is a cornerstone in ensuring the acceptance of the recommendations by healthcare professionals and patients alike.
